# Recommendations for 24-Hour Movement Behaviours in Adults with Asthma: A Review of Current Guidelines

**DOI:** 10.3390/ijerph17051789

**Published:** 2020-03-10

**Authors:** Shilpa Dogra, Ilana Patlan, Carley O’Neill, Hayley Lewthwaite

**Affiliations:** 1Faculty of Health Sciences (Kinesiology), University of Ontario Institute of Technology, Oshawa, ON L1G 0C5, Canada; Ilana.Patlan@ontariotechu.net (I.P.); Carley.Oneill@ontariotechu.net (C.O.); 2Department of Kinesiology and Physical Education, McGill University, Montreal, QC H2W 1S4, Canada; Hayley.Lewthwaite@mcgill.ca; 3Innovation, Implementation and Clinical Translation in Health (IIMPACT), School of Health Sciences, University of South Australia, Adelaide 5001, Australia

**Keywords:** exercise, physical activity, sedentary, sleep, lung, respiratory

## Abstract

*Background*: Many countries have clinical practice guidelines (CPG) for asthma that serve as an important resource for healthcare professionals and inform the development of policies and practices relevant to asthma care. The purpose of this scoping review was to search for CPGs related to asthma to determine what recommendations related to the 24-h movement behaviours are provided. *Methods*: We searched for the most recent CPGs published by a national authoritative body from 195 countries. Guidelines were reviewed for all movement behaviours; that is, physical activity, sedentary behaviour, and sleep. *Results*: In total, 82 documents were searched for eligibility and 19 were included in our review. Of these, only 10 CPGs provided information on physical activity; none provided recommendations consistent with the FITT principle, while seven recommended activity levels similar to the general population. None of the guidelines included information on sedentary behaviour. Nine guidelines included information on sleep: recommendations mostly focused on changes to medication to reduce disruptions in sleep. *Conclusions*: It is recommended that future work be conducted to create comprehensive movement behaviour guidelines accompanied with relevant precautions and strategies to ensure that adults with asthma are able to safely and effectively engage in movement behaviours throughout the day.

## 1. Introduction

There is unequivocal evidence suggesting that regularly engaging in physical activity is associated with better health and clinical outcomes in adults with asthma [1,2,3]. Yet, research indicates that over 50% of adults with asthma are inactive, [4] and those with severe asthma are less active than their age-matched peers without asthma [5]. Furthermore, adults with asthma are spending a significant amount of time in sedentary behaviours [2,6]. Sedentary behaviour is defined as any activity performed in the seated or reclined posture that requires low energy expenditure [7]. It is distinct from physical activity. For example, some individuals may participate in sports or exercise regularly but also watch several hours of television per day. In fact, research is now emphasising the full 24 h of movement instead of focusing on distinct movement behaviours. Specifically, each 24-h day, or 24-h activity cycle, is spent in either sedentary time, light intensity physical activity, moderate-to-vigorous intensity physical activity, or sleep [8]. Research suggests that it is the composition of the 24 h that is more important for health outcomes, rather than any one of these movement behaviours alone [3,9,10]. This model also appreciates the inter-relationship between the four movement behaviours, and their combined effects on health outcomes [8]. Countries are shifting away from the creation of physical activity guidelines and moving towards the creation of 24-h movement guidelines [11,12]. In such guidelines, the focus is on all four movements, and attaining an adequate amount of each behaviour, instead of simply focusing on moderate-to-vigorous intensity physical activity. 

The current recommendation for healthy adults and older adults is to accumulate a minimum of 150 min of moderate-to-vigorous intensity physical activity per week and to engage in strength training activities at least twice per week [13]. Several clinical practice guidelines have adapted these recommendations to align with specific clinical populations based on available evidence. For example, the Diabetes Canada Clinical Practice Guidelines contains a chapter entitled “Physical Activity and Diabetes” in which the expert committee reviews the evidence and provides specific recommendations relevant to those with diabetes [14]. Of particular importance, such guidelines include information on the recommended dose of physical activity as well as recommendations relevant to safety while performing different intensities or types of physical activity. Such guidelines are critical for clinicians and adults with chronic conditions to ensure that firstly, clinicians are able to effectively and confidently prescribe exercise or provide guidance with regards to 24-h movement behaviours, and secondly, to ensure that adults with chronic conditions are able to safely and effectively adopt an active lifestyle.

Among adults with asthma, recommendations pertaining to safety as well as dose of physical activity are particularly important due to the effect that different intensities of exercise have on triggering exercise-induced bronchoconstriction (EIBC). Based on the pathophysiology of EIBC, exercise associated with higher ventilation [15] and performed in the presence of other triggers may increase the frequency and/or severity of EIBC [16]. For example, exercise in cold, dry air, or in the presence of higher air pollution or other allergens, exacerbates EIBC [17,18]. Thus, additional recommendations would be needed with regards to medication use [19], warm-up [20], and avoidance of other triggers to prevent and manage EIBC symptoms. Similarly, recommendations with regards to screen time and sleep might be crafted to target adults with asthma. For example, recommendations regarding sleep hygiene and nighttime symptoms could be provided to ensure adults are able to obtain the recommended 7–9 h of sleep per night [21,22]. 

It is imperative that adults with asthma be given 24-h movement behaviour recommendations that carefully consider the unique characteristics of their condition. The purpose of this scoping review was to search for guidelines related to asthma to determine what recommendations related to any of the 24-h movement behaviours are provided, and to determine whether there is consensus in these recommendations. Specifically, this scoping review aimed to answer two research questions regarding clinical practice guidelines for the management of asthma in adults:What is recommended around specific patterns, volumes, types or frequency of physical activity, sedentary behaviour or sleep?What strategies are recommended to optimise physical activity, sedentary behaviour or sleep?

Given that the prevalence of asthma may be upwards of 20% in some countries [23], and an active lifestyle is associated with better health in adults with asthma, such recommendations are immediately needed. 

## 2. Materials and Methods 

We undertook a scoping review of asthma clinical practice guidelines (CPG) from all countries (n = 195) across the world. While this was not a systematic review of the published literature, we followed the PRISMA guidelines to structure our methods and reporting [24]. 

***Eligibility***: Guidelines were eligible for inclusion in this review if they: (1) were the most recent version of a national CPG for the management of asthma in adults; (2) were developed by an authoritative scientific, medical or government body by review of the relevant literature of scientific/medical consensus; (3) mentioned specific terms relating to physical activity, exercise, sedentary behaviour or sleep (“physical activity”, “exercise*”, “cycle”, “bike”, “walk”, “strength”, “weight”, “resistance”, “endurance”, “stand”, “sit”, “screen”, “yoga”, “sleep”, “rest”, and “sedentary”); and (4) provided recommendations unique from the Global Initiative for Asthma (GINA) guidelines; that is, if guidelines only reiterated recommendations provided within GINA then they were not included. Guidelines included were limited to the English language. References were excluded if they were original research articles, opinion pieces or reviews reporting on primary research data. 

***Information sources and search strategy***: Google search engine was used to identify eligible reviews. The decision to use Google over traditional electronic databases that contain peer-reviewed literature was based on a previous review of CPGs on COPD, which indicated that most clinical practice guidelines are not found in the published literature and are therefore not indexed in electronic databases [25]. Individual search strings were created for each of the 195 countries across the world, which comprised the terms “asthma” (population of interest), “guideline” or “recommendation” or “management” (publication type), and the country of interest (e.g., “Canada”); thus, three search terms per country. Search results on the first three pages (thus nine pages in total for each country) were considered. Links were investigated if they referred to a guideline and were followed until guidelines were found. The websites of known international authoritative bodies (Global Initiative for Asthma, World Health Organization, Global Asthma Network) were also searched for available national guidelines. Finally, we used the Global Asthma Report [26] and the Global Asthma Network [27] to identify additional guidelines.

***Study selection***: References identified were downloaded as full text and screened against eligibility criteria by two independent reviewers (CO, IP). Each guideline was searched for relevant search terms relating to movement behaviours; that is, sedentary behaviour, light intensity physical activity, moderate-to-vigorous intensity physical activity, and sleep. 

***Data collection and items***: A data extraction template was developed *a priori* and two independent reviewers (CO, IP) extracted data from eligible guidelines on:-Publication demographics: title, country of development, authoritative development body, year of publication, target age group (adult or adult and paediatric);-Recommendations around movement behaviours: whether recommendations were provided around movement behaviours (i.e., around the frequency, intensity, type, volume or context of physical activity, sedentary behaviour or sleep as well as recommendations pertaining to medication use) (yes/no), recommendations verbatim, whether a strategy to achieve movement behaviour recommendations was provided (i.e., a specific method or means to achieving a change in a movement behaviour) (yes/no), recommended strategy verbatim; and-Whether consumers were involved in the development of guidelines (yes/no).

Data extracted were compared between reviewers for conflicts with inconsistencies resolved by discussion and consensus. If required, a third reviewer (HL, SD) was consulted. 

***Data Analysis***: Data extracted from CPGs were collated and summarised descriptively by movement behaviour type. Recommendations around movement behaviours and strategies to achieve improvements in movement behaviours were compared for commonalities across guidelines. 

## 3. Results

There were 82 references identified in the initial search. After removal of duplicates, 59 unique guidelines remained; articles were removed because they were not in English (n = 10), were not a national guideline (n = 22) or did not include relevant terms (n = 8) (Figure 1). For three guidelines, authors were contacted for full-text articles, no response was received prior to the initial submission of the manuscript. In total, 19 CPGs were included in the final review. Guidelines were published between 2005 and 2019 from 19 unique countries, specifically targeting adults only (n = 8) [28,29,30,31,32,33,34,35] or adults and children (n = 11) [36,37,38,39,40,41,42,43,44,45,46]. The recommendations and strategies provided for each movement behaviour are described below.

***Physical activity:*** Of the 19 included CPGs, 10 (53%) provided a recommendation around physical activity (Table 1). No guideline provided recommendations consistent with the ‘FITT’ principle [13]; that is, recommending a specific frequency, intensity, time and type of physical activity. Seven (37%) CPGs recommended a general frequency or dose of physical activity: normal/near normal levels [32,34,35,38,41] or regular levels [30,36]. The Australian CPG was the sole guideline to recommend an intensity, recommending ‘moderately intense’ physical activity [36]. Three guidelines provided recommendations around the type of physical activity. The India and UK CPGs recommended yoga or relaxation exercises [45,46]; the Australian guideline recommended weight bearing or resistance exercise for people with asthma taking oral corticosteroids [36]; and the UK guideline recommended physical training to improve cardiopulmonary efficiency, or weight loss programs for people with asthma who are overweight or obese [45]. The Iraq and Swiss guidelines provided a non-specific recommendation of ‘physical activity or exercise’ for asthma management [34,43].

In total, 16 (84%) CPGs recommended strategies for how to improve/maintain levels of physical activity (Table 1). The most common strategy, recommended by 14 (74%) CPGs, was the administration of short-acting β2-agonist prior to physical activity or exercise participation. The Australian CPG was the most specific in its recommendation, recommending 1–4 puffs via a metered dose inhaler 15 min before exercise [36]. Other CPGs recommended 10–15 min prior to exercise [30,42], 15–20 min prior to exercise [29,41], immediately/shortly before exercise [32,40,45] or no specific dose or timing. The next most common recommended strategies were: advice/support/encouragement from the healthcare provider (n = 12, 63%), long-acting preventative medication (n = 10, 53%), avoiding environmental triggers such as cold/dry weather and high pollution (n = 7, 37%) and adequate warm-up and cool-down before/after exercise (n = 6, 32%). Verbatim recommendations and strategies around physical activity are presented in Appendix A.

***Sedentary behaviour:*** No CPG provided recommendations around sedentary behaviour. In fact, the term sedentary behaviour, or any terms relating to sedentary behaviour, was only mentioned in a single guideline by the United States [32]. The mention of sedentary behaviour was related to an increased risk of low bone mineral content in sedentary older adults with asthma taking inhaled corticosteroids [32]. 

***Sleep:*** Of the nine CPGs that included information pertaining to sleep, none provided recommendations around sleep patterns or volumes for people with asthma. Two CPGs provided recommendations around improving the sleep environment to facilitate sleep/reduce asthma symptoms during sleep, including removing items from the bedroom that accumulate dust, and pets [28,32]. Eight of the CPGs recommended changes to pharmaceutical treatment if patients were experiencing sleep disturbances; sleep was used as an indicator of asthma control [29,30,32,34,36,41,45,46]. Verbatim recommendations and strategies around sleep are presented in Appendix A.

Of the 19 guidelines included, nine performed critical appraisal and provided levels of evidence for their recommendations. One of the CPGs did not provide specific physical activity or sleep recommendations [39], four CPGs included levels of evidence for drugs and not physical activity or sleep recommendations [30,35,40,43], and the remaining CPGs (n = 4) provided some grading or level of evidence for some or all of the physical activity and sleep related recommendations [32,41,42,45]; evidence levels are included with the recommendations in the Appendix A. Evidence levels varied such that recommendations and strategies pertaining to drugs had higher evidence level ratings than those pertaining to behaviour (e.g., cover your mouth with a mask). None of the guidelines included patients as part of their review based on description of methodology, authorship or acknowledgement sections in the guidelines. 

## 4. Discussion

Asthma affects a large proportion of the adult population worldwide [23]. An active lifestyle and adequate sleep are associated with better asthma control, quality of life, and clinical outcomes [3,47,48]. We sought to review recommendations provided to adults with asthma with regards to 24-h movement behaviours to determine gaps and to identify avenues for future work. With regards to physical activity, no guidelines provided specific recommendations around frequency, intensity, time and type of physical activity. None of the guidelines reviewed provided recommendations regarding sedentary time, and while some provided recommendations pertaining to sleep, these lacked specificity. Together, these findings suggest significant gaps in current national CPGs. 

While we only included national level guidelines in our search, it must be acknowledged that the GINA guidelines [27] are an important reference for healthcare professionals working with individuals with asthma. Not surprisingly, the GINA guidelines do not provide specific recommendations around physical activity, recommending ‘regular moderate physical activity’, and no recommendations are provided around sedentary or sleep patterns/volumes. Strategies recommended to increase or maintain levels of physical activity include maintenance therapy with inhaled corticosteroids, the use of short-acting bronchodilators or low-dose inhaled corticosteroids before/during exercise, warm-up prior to exercise, encouragement from the healthcare provider and avoidance of unfavorable environmental conditions. Some strategies are provided by GINA around improving sleep pertaining to the control of allergens in the bedroom that would be associated with nighttime symptoms. Like the findings of the current review of national guidelines, however, the recommendations and strategies provided by GINA around movement behaviours could be further improved. 

### 4.1. Physical Activity

This review covered physical activity of both light and moderate-to-vigorous intensity and used exercise-related terms to search for information relevant to recommendations. While the definition and classification of asthma control considers limitations to physical activity [49], the guidelines reviewed did not provide information on how to counter the limitations experienced or how to increase levels of physical activity in daily life. This is problematic as limited activity can quickly lead to deconditioning, which, in turn, may exacerbate asthma-related risks [50]. As previously mentioned, asthma is somewhat unique in that the increase in ventilation that accompanies physical activity, especially higher intensities of aerobic physical activity, often leads to acute bronchoconstriction [15]. As such, any adult with asthma should be provided guidance on how to safely warm-up, how to use medications prophylactically or during acute bronchoconstriction, how to avoid additional triggers such as exposure to cold, dry air (e.g., using a mask), or exposure to outdoor allergens (e.g., by exercising indoors). These simple strategies can reduce the frequency and severity of exercise-induced symptoms, allowing adults with asthma to exercise safely. It would also reduce the negative effects associated with exercise and could lead to better long-term adherence to an active lifestyle [51]. 

With regards to the dose of physical activity, current recommendations for adults in the general population are likely applicable [13]; however, several recommendations could be provided to adults with asthma that are unique to this population. For example, interval exercise has been shown to be associated with an attenuated airway response [52] and thus may be a better alternative to continuous exercise. Furthermore, the mode of exercise may be important; swimming in an indoor pool provides a warm, humid environment, and resistance training does not lead to the same increase in ventilation as aerobic exercise, particularly in beginners [53,54,55]. 

There are many benefits to participating in regular physical activity for adults with asthma [50]. However, rates of participation remain low. While some healthcare professionals may be providing guidance to patients, there are no guidelines that currently address the type of information or the dose of physical activity recommended. This is a significant gap in asthma care, as many young adults with asthma do not receive supervised exercise in rehabilitation settings like older adults in pulmonary or cardiac rehabilitation. 

### 4.2. Sedentary Time

In general, there is a dearth of research on sedentary time in adults with asthma. However, existing research clearly indicates that a higher sedentary time is associated with worse asthma-related outcomes [2,3]. Furthermore, for general health and mortality, a large body of evidence indicates that reducing sedentary time is critically important [56]. Nevertheless, none of the guidelines reviewed addressed sedentary time or the impact that prolonged sedentary time may have on one’s asthma or general health. In the absence of high-quality evidence in asthma, leaders in the field should consider looking to research on healthy adults and using clinical expertise to provide guidelines pertaining to sedentary behaviour. As with healthy adults, replacing sedentary time with light intensity physical activity may be a better starting point for adopting an active lifestyle than engaging in moderate-to-vigorous intensity physical activity. Reducing sedentary time may also influence sleep quality. Thus, the additive effect of engaging in all movement behaviours optimally may have an even greater effect on asthma outcomes.

### 4.3. Sleep

Nighttime symptoms and sleep disruptions are an important indicator of asthma control [49]. Surprisingly, few guidelines provided recommendations or strategies to improve sleep quality and minimise nighttime symptoms. Only two studies provided recommendations pertaining to sleep hygiene and allergens, while eight CPGs commented on medication to minimise sleep disruptions. Importantly, none of the guidelines recommend how many hours of sleep should be attained, or what to do in case of sleep disruptions. Given the strong association between sleep and health [57], and the effect that poorly controlled asthma can have on sleep, this is a topic that should be addressed. 

### 4.4. Methodological Considerations 

It is important to note that our search method and search engine did not capture all of the CPGs available on asthma, particularly those that are not available in English. Nevertheless, we were able to find guidelines for 82 countries of the 195 searched. Based on previous research on guidelines, our method appears to be more appropriate for identifying guidelines than a review of databases containing peer-reviewed literature. Finally, none of the guidelines reviewed included patients in their development process. Future guidelines committees may want to consider inclusion of patients, as this may inform the inclusion of additional information, particularly pertaining to movement behaviours. 

### 4.5. Implications and Future Directions 

It is clear from the review of the current CPGs available on asthma that movement behaviours are not fully considered. In some cases, this may be due to a lack of research available to inform such guidelines. High quality data on 24-h movement behaviours are needed to better understand dose–response associations between each movement behaviour and asthma-relevant outcomes such as asthma control and asthma-related quality of life. Future research may also want to carefully consider appropriate behaviour-change techniques and counseling strategies to help adults with asthma overcome their fear of exercise-induced symptoms and to adopt and maintain an active lifestyle. In particular, research on long-term behaviour change using a stepwise approach, starting from sedentary behaviour and moving towards moderate-to-vigorous intensity physical activity over time, should be examined. Together, these areas of research can better inform the development of asthma-specific, evidence-based recommendations for doses of movement behaviours, as well as the strategies to achieve these recommendations. However, in the meantime, where there is lack of asthma-specific evidence, CPGs should work to include recommendations and strategies around movement behaviours informed by what is known in the general population and expert consensus. This would likely require collaborative efforts across the fields of asthma clinical management and 24-h integrated movement behaviors. 

## 5. Conclusions

In conclusion, we found that CPGs from the select countries that we identified and reviewed did not provide comprehensive recommendations for physical activity or sleep, or address the issue of sedentary behaviour. Given the importance of activity and sleep in the proper management of asthma, experts in the area are urged to consider movement behaviours in forthcoming guidelines. Collaboration with experts in physical activity, sedentary behaviour, and sleep can allow for the development of appropriate recommendations. 

## Figures and Tables

**Figure 1 ijerph-17-01789-f001:**
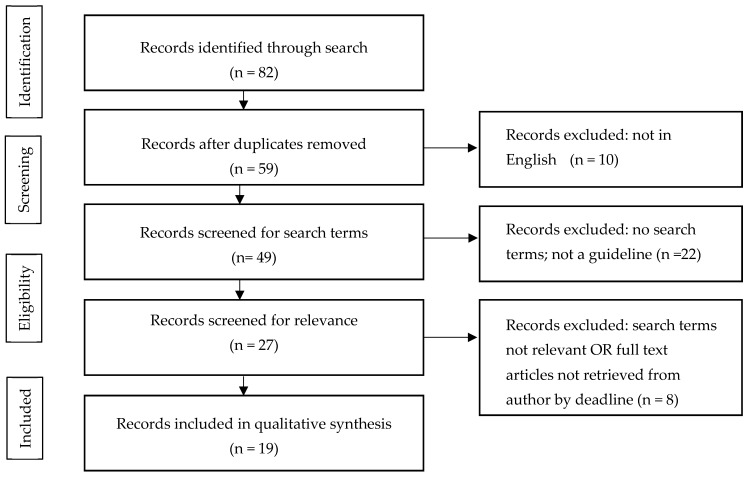
Flow chart for number of guidelines identified, screened, assessed for eligibility, and included in review.

**Table 1 ijerph-17-01789-t001:** Physical Activity Recommendations and Strategies from Clinical Practice Guidelines on Asthma.

CPG	Recommendations				Strategies								
	F	I	Time	Type	HCP Advice/Support	Patient Action/Edu	Short-Acting Meds	Long-Acting Meds	Warm-Up/Cool Down	Avoid Enviro Trigger	Mask over Face	Diet	Referral
India, 2005	-	-	-	Yoga/relaxation									
Japan, 2014	-	-	-	-									
Iraq, 2012	Normal/near	-	-	Exercise/other PA									
Australia, 2019	Regular	Mod.	-	Weight bear/RT *									
Malaysia, 2017	-	-	-	-									
NZ, 2016	-	-	-	-									
Oman, 2009	Normal	-	-	-									
Qatar, 2016	Regular	-	-	-									
Saudi Arabia, 2019	-	-	-	-									
S. Africa, 2007	Normal	-	-	-									
Spain, 2016	-	-	-	-									
Switzerland, 2018		-	-	PA									
Turkey, 2011		-	-	-									
UK, 2019	-	-	-	Yoga/PT/weight loss									
USA, 2007	Normal	-	-	-									
Caribbean, 2009	-	-	-	-									
Ireland, 2013	Normal	-	-	-									

* for people at risk of osteoporosis; F: frequency; I: Intensity; NS: non-specific; RT: resistant training; PT: physical training; HCP: Healthcare professional. Shaded boxes indicate that information on the strategy was included.

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
