# Peer review of "Recommendations for 24-Hour Movement Behaviours in Adults with Asthma: A Review of Current Guidelines"

_ijerph, 2020, doi:10.3390/ijerph17051789_

Round 1

Reviewer 1 Report

Introduction

Line 73- please give the specific research questions that were framed based on the purpose of the review.

Methods

It would be good for the authors to mention why they used the Google search engine as their main source and not others. Also, acknowledge the limitations of using Google. What were the criteria they used to filter reliable sources of information versus others? Make it clear what the authors mean by movement behaviors. What were the behaviors they were looking at? Why did they decide to focus on these movement behaviors versus others?

Results and discussion

Please mention the other reasons (other than no response from the authors) in the text why out of 59 articles, only 19 were included-; line 124 It would be good for the authors to assess the risk of bias and/or limitations of the guidelines studies that they included concisely in the table.
And have a results section or overview of these limitations.

Other comments

It would be good for the authors to have a section on implications for research since clearly there are gaps to be filled in several areas and the existing guidelines are not detailed.

Author Response

Comment: Introduction: Line 73- please give the specific research questions that were framed based on the purpose of the review.

Response: We have now added specific research questions for this review. The review aim and research questions now read:

“The purpose of this scoping review was to search for guidelines related to asthma to determine what recommendations related to any of the 24-hour movement behaviours are provided, and to determine whether there is consensus in these recommendations.

Specifically, this scoping review aimed to answer two research questions regarding clinical practice guidelines for the management of asthma in adults:

  1. What is recommended around patterns, volumes, type or frequency of physical activity, sedentary behaviour or sleep?; and
  2. What strategies are recommended to optimise physical activity, sedentary behaviour or sleep?

Comment: It would be good for the authors to mention why they used the Google search engine as their main source and not others. Also, acknowledge the limitations of using Google. What were the criteria they used to filter reliable sources of information versus others?

Response: Thank you for raising these questions, as this information is certainly important for the reader. Our choice to use Google was primarily based on a similar review of guidelines for COPD that was previously published by one of the co-authors (Lewthwaite et al. PMID: 5720236). This review was a systematic review, following PRISMA guidelines, which searched for guidelines using electronic databases. The authors of this review found that clinical practice guidelines are rarely published in scientific journals (especially more recent versions) and are therefore not index to electronic databases. Of the 35 guidelines included in this review, 22 (63%) were identified by other sources.

“The decision to use Google over traditional electronic databases that contain peer-reviewed literature was based on a previous review of clinical practice guidelines on COPD which indicated that most clinical practice guidelines are not found in published literature and are therefore not indexed in electronic databases [25].”

The reviewer is absolutely correct in pointing out that using this search engine was a limitation. We have now included the following line in our discussion section:

“It is important to note that our search method and search engine did not capture all of the CPGs available on asthma, particularly those that are not available in English. Nevertheless, we were able to find guidelines for 82 countries of the 195 searched.”

The criteria used while searching was simply whether the link provided a “guideline” or linked to a document. This step has been further clarified in the methods section as well.

“An individual search string was created for each of the 195 countries across the world, which comprised the terms “asthma” (population of interest), “guideline” or “recommendation” or “management” (publication type), and the country of interest (e.g., “Canada”); thus, three searches per country. Search results on the first three pages (thus 9 pages in total for each country) were considered. Links were investigated if they referred to a guideline, and were followed until guidelines were found.”

Comment: Make it clear what the authors mean by movement behaviors. What were the behaviors they were looking at? Why did they decide to focus on these movement behaviors versus others?

Response: Thank you for this suggestion. We agree with you and reviewer 2 that additional information is needed to clarify what we mean by ‘movement behaviours’. We chose to categorises daily activities/behaviours by their intensity; a common approach used in activity research as well as the method commonly used in movement behaviour/physical activity guidelines.

We have added information to the introduction, and have further detailed the line mentioned here:

“Each guideline was searched for relevant search terms relating to movement behaviours, that is, sedentary behaviour, light intensity physical activity, moderate to vigorous intensity physical activity, and sleep.”

Comment: Please mention the other reasons (other than no response from the authors) in the text why out of 59 articles, only 19 were included-;

Response: We have now included the following line in text:

“After removal of duplicates, 59 unique guidelines remained; articles were removed because they were not in English (n=10), were not a national guideline (n=22), did not include relevant terms (n=8).”

Comment: line 124 It would be good for the authors to assess the risk of bias and/or limitations of the guidelines studies that they included concisely in the table. And have a results section or overview of these limitations.

Response: Thank you for this comment. We agree that risk of bias is important to address in reviews. While some of the guidelines included in this manuscript based recommendations on original research, many did not, and rather recommendations may be based on consensus expert opinion. The purpose of this review was not to appraise the quality of the guidelines or the quality of the evidence used to inform the recommendations. Rather, the purpose of this review was to state what was recommended by the guideline. 

For transparency of how the guidelines were developed, we have added the following line to indicate the number of guidelines that conducted their own appraisal of the research. The added line reads as follows:

“Of the 19 guidelines included, nine performed critical appraisal and provided level of evidence for their recommendations.”

Comment: It would be good for the authors to have a section on implications for research since clearly there are gaps to be filled in several areas and the existing guidelines are not detailed.

Response: This is a great suggestion, thank you. We have added detail to our future research section. It now reads as follows:
“It is clear from the review of the current CPGs available on asthma that movement behaviours are not fully considered. In some cases, this may be due to a lack of research available to inform such guidelines. High quality data on 24-hour movement behaviours is needed to better understand dose response associations between each movement behaviour and asthma-relevant outcomes such as asthma control and asthma related quality of life. Future research may also want to carefully consider appropriate behaviour change techniques and counseling strategies to help adults with asthma overcome their fear of exercise-induced symptoms and to adopt and maintain an active lifestyle. In particular, research on long-term behaviour change using a step-wise approach, starting from sedentary behaviour and moving towards moderate-vigorous intensity physical activity over time, should be examined. Together, these areas of research can inform the development of evidence based recommendations for doses of movement behaviours, as well as the strategies to achieve the recommendations.”

Reviewer 2 Report

 This manuscript is relevant and pertinent. It reads rather well. I suggest minor essential changes:

INTRODUCTION
The introduction frames well the study, however, the concept of 24 hour movement behaviour should be better addressed and defined; some papers use the “an integration of physical activity, sedentary behavior, and sleep”, but this should be explained. Note: many clinicians may not be aware of this concept.
METHODS: Since authors report that only guidelines in English language were included, it would be pertinent to describe which countries were excluded.
RESULTS/DISCUSSION
Authors should briefly describe the study sample, i.e the set of countries whose CPG were reviewed.
Did any guideline engage asthma patients’ representatives? This should be taken into account in the analysis; patient involvement should be discussed in the discussion section and conclusions/implications.
Authors should acknowledge study limitations, in particular considering the great number of CPG excluded due to non-English language.

CONCLUSIONS
Authos write “In conclusion, we found that current CPGs of most countries do not provide comprehensive….”
To use most countries is a forced generalization taken from a non-representative sample. Authors should be more cautious about this.

ABSTRACT

Change abstract accordingly to requested changes. 

Author Response

Comment: This manuscript is relevant and pertinent. It reads rather well. I suggest minor essential changes:

Response: Thank you for your positive response.

Comment: The introduction frames well the study, however, the concept of 24 hour movement behaviour should be better addressed and defined; some papers use the “an integration of physical activity, sedentary behavior, and sleep”, but this should be explained. Note: many clinicians may not be aware of this concept.

Response: The reviewer brings up an excellent point about familiarity of the 24 hour movement behaviours among clinicians. We have expanded on movement behaviours in the introduction.  Thank you for this suggestion.

Comment: Since authors report that only guidelines in English language were included, it would be pertinent to describe which countries were excluded.

Response: As per our PRISMA flow diagram, ten guidelines were removed because they were not available in English. Furthermore, our search was conducted in English, which likely limited the results. Of the 195 countries searched, we found 82 results (42%). We have added the following line to the manuscript to address this limitation:

“It is important to note that our search method and search engine did not capture all of the CPGs available on asthma, particularly those that are not available in English. Nevertheless, we were able to find guidelines for 82 countries of the 195 searched.”

Comment: Authors should briefly describe the study sample, i.e. the set of countries whose CPG were reviewed. Did any guideline engage asthma patients’ representatives? This should be taken into account in the analysis; patient involvement should be discussed in the discussion section and conclusions/implications.
Response: We agree with the reviewer that patient involvement is important and likely would have influenced the information included in the guidelines. We have added the following lines to the manuscript to address this:

“None included patients as part of their review based on authorship and acknowledgement sections in the guidelines.”

“Finally, none of the guidelines reviewed included patients in their development process. Future guidelines committees may want to consider inclusion of patients, as it may inform inclusion of additional information, particularly pertaining to movement behaviours.”

Comment: Authors should acknowledge study limitations, in particular considering the great number of CPG excluded due to non-English language.

Response: We completely agree with the reviewer, and have added the following line to the discussion section of the manuscript:

“It is important to note that our search method and search engine did not capture all of the CPGs available on asthma, particularly those that are not available in English. Nevertheless, we were able to find guidelines for 82 countries of the 195 searched.”

Comment: Authors write “In conclusion, we found that current CPGs of most countries do not provide comprehensive….” To use most countries is a forced generalization taken from a non-representative sample. Authors should be more cautious about this.

Response: This is a good point. We have edited this line to better reflect the representativeness of this sample. The line now reads:

“In conclusion, we found that CPGs from the select countries that we identified and reviewed did  not provide comprehensive recommendations for physical activity or sleep, or address the issue of sedentary behaviour..”

Comment: Change abstract accordingly to requested changes. 

Response: We have made changes in the methods to reflect the reviewer’s comments.

Round 2

Reviewer 1 Report

The authors did a nice job of addressing the comments. The manuscript reads much better.

I just have one more suggestion based on one of my previous comments. Would the authors please expand on this paragraph "Of the 19 guidelines included, nine performed critical appraisal and provided levels of evidence for
190 their recommendations. None included patients as part of their review based on authorship and 191 acknowledgement sections in the guidelines."?

How did the nine articles perform critical appraisal? What were the strengths and limitations that they noted?

For these major limitations, please include next steps for research (in the section of implications for research) if those have not yet been incorporated in the review.

Author Response

Comment: The authors did a nice job of addressing the comments. The manuscript reads much better.

Response: Thank you!

Comment: I just have one more suggestion based on one of my previous comments. Would the authors please expand on this paragraph "Of the 19 guidelines included, nine performed critical appraisal and provided levels of evidence for their recommendations. None included patients as part of their review based on authorship and acknowledgement sections in the guidelines."? How did the nine articles perform critical appraisal? What were the strengths and limitations that they noted?

Response: This is a good point, thank you for raising it. We have now added the following line to the manuscript as well as accompanying information in the supplementary tables:

“Of the 19 guidelines included, nine performed critical appraisal and provided levels of evidence for their recommendations. One of the CPGs did not provide specific physical activity or sleep recommendations [39], four CPGs included levels of evidence for drugs and not physical activity or sleep recommendations [30, 35, 40, 43], and the remaining CPGs (n=4) provided some grading or level of evidence for some or all of the physical activity and sleep related recommendations [32, 41, 42, 45]; evidence levels are included with the recommendations in the supplementary tables. Evidence levels varied such that recommendations and strategies pertaining to drugs had higher evidence level ratings than those pertaining to behaviour (e.g. cover your mouth with a mask). None of the guidelines included patients as part of their review based on description of methodology, authorship or acknowledgement sections in the guidelines.”

Comment: For these major limitations, please include next steps for research (in the section of implications for research) if those have not yet been incorporated in the review.

Response: Thank you for this suggestion. To highlight the implications of our review and future directions, we have added a heading to the discussion ‘Implications and future directions” and have expanded the section to include further clear information on the next steps forward in this area. This section now reads as below, with new text in red:

Implications and future directions

It is clear from the review of the current CPGs available on asthma that movement behaviours are not fully considered. In some cases, this may be due to a lack of research available to inform such guidelines. High quality data on 24-hour movement behaviours are needed to better understand dose-response associations between each movement behaviour and asthma-relevant outcomes such as asthma control and asthma-related quality of life. Future research may also want to carefully consider appropriate behaviour change techniques and counseling strategies to help adults with asthma overcome their fear of exercise-induced symptoms and to adopt and maintain an active lifestyle. In particular, research on long-term behaviour change using a stepwise approach, starting from sedentary behaviour and moving towards moderate-vigorous intensity physical activity over time, should be examined. Together, these areas of research can better inform the development of asthma-specific evidence-based recommendations for doses of movement behaviours, as well as the strategies to achieve the recommendations. However, in the meantime, where there is lack of asthma-specific evidence, CPGs should work to include recommendations and strategies around movement behaviours informed by what is known in the general population and expert consensus. This would likely require collaborative efforts across the fields of asthma clinical management and 24-hr integrated movement behaviors.